# Developing a Hybrid Neuro-Fuzzy Method to Predict Carbon Dioxide (CO_2_) Permeability in Mixed Matrix Membranes Containing SAPO-34 Zeolite

**DOI:** 10.3390/membranes12111147

**Published:** 2022-11-16

**Authors:** Ali Hosin Alibak, Seyed Mehdi Alizadeh, Shaghayegh Davodi Monjezi, As’ad Alizadeh, Falah Alobaid, Babak Aghel

**Affiliations:** 1Chemical Engineering Department, Faculty of Engineering, Soran University, Soran 44008, Iraq; 2Petroleum Engineering Department, Australian University, West Mishref 11411, Kuwait; 3Department of Environmental Science, Faculty of Natural Resources and Marine Science, Tarbiat Modares University, Nur 46414356, Iran; 4Department of Civil Engineering, College of Engineering, Cihan University-Erbil, Erbil 44001, Iraq; 5Institut Energiesysteme und Energietechnik, Technische Universität Darmstadt, Otto-Berndt-Straße 2, 64287 Darmstadt, Germany; 6Department of Chemical Engineering, Faculty of Energy, Kermanshah University of Technology, Kermanshah 6715685420, Iran

**Keywords:** mixed matrix membrane, SAPO-34 zeolite, carbon dioxide separation, theoretical analysis, adaptive neuro-fuzzy inference system (ANFIS)

## Abstract

This study compares the predictive performance of different classes of adaptive neuro-fuzzy inference systems (ANFIS) in predicting the permeability of carbon dioxide (CO_2_) in mixed matrix membrane (MMM) containing the SAPO-34 zeolite. The hybrid neuro-fuzzy technique uses the MMM chemistry, pressure, and temperature to estimate CO_2_ permeability. Indeed, grid partitioning (GP), fuzzy C-means (FCM), and subtractive clustering (SC) strategies are used to divide the input space of ANFIS. Statistical analyses compare the performance of these strategies, and the spider graph technique selects the best one. As a result of the prediction of more than 100 experimental samples, the ANFIS with the subtractive clustering method shows better accuracy than the other classes. The hybrid optimization algorithm and cluster radius = 0.55 are the best hyperparameters of this ANFIS model. This neuro-fuzzy model predicts the experimental database with an absolute average relative deviation (AARD) of less than 3% and a correlation of determination higher than 0.995. Such an intelligent model is not only straightforward but also helps to find the best MMM chemistry and operating conditions to maximize CO_2_ separation.

## 1. Introduction

Environmental pollution [1], global warming [2], and climate change [3], are likely the most interconnected problematic issues in recent decades. Carbon dioxide (CO_2_) as a greenhouse gas is undeniably a key factor in creating these undesirable environmental phenomena [4]. Therefore, separating and recycling CO_2_ from the gaseous streams released to the atmosphere seems essential. In addition, separating the CO_2_ impurity of the natural gas is significant to improve the heating value and avoid corrosion in transmission pipelines and processing equipment. Absorption [5], adsorption [6,7], cryogenic process [8,9], and membrane-based [10] operations are conventionally applied to separate CO_2_ molecules from either waste or raw gaseous stream. In addition, the recovered carbon dioxide can be then used for the separation of heavy metal pollutants [11]. 

The membrane-based technologies, which are low cost, highly flexible, easy to operate, energy saving, and environmentally friendly [12], have seen a growing interest in the field of CO_2_ capture. Due to their suitable mechanical characteristics, the polymeric membranes work better in high-pressure and temperature situations. Researchers have continuously attempted to fabricate materials that fulfill the trade-off between gas selectivity and permeability and overcome the ‘‘Robeson upper bond’’ curve in polymeric membranes. To do so, attention has been concentrated on the inclusion of inorganic solid materials in the polymer membrane structure [13].

The mixed matrix membranes (MMM), which combine a polymer as the continuous phase and inorganic filler as the dispersed phase, have been suggested to resolve these limitations [14]. This special class of membranes shows better gas separation performance than both polymer and inorganic fillers [15]. Activated carbon [16], nano-fibers [17], nanoparticles [18], metal-[19] and [20] covalent organic frameworks, and zeolites [21], have been successfully dispersed in polymers’ structure to fabricate MMMs.

Due to high pore volume, uniform pore size distribution, and appropriate chemical/thermal stability, zeolites [22] and their inclusion in a matrix [23] are often used for the CO_2_ separation. The SAPO-34 zeolite, with excellent CO_2_ adsorption affinity, has achieved great popularity in synthesizing mixed matrix membranes [23]. These investigations have mainly measured the CO_2_ permeability in mixed matrix membranes experimentally. The permeability measurements have been considered as a function of polymer type, SAPO-34 dosage, pressure, and temperature. 

Despite a relatively extensive experimental analysis of CO_2_ permeability in MMMs, there is no systematic strategy to predict this key variable. Therefore, the current research uses a hybrid neuro-fuzzy modeling approach to predict CO_2_ permeability in MMMs containing SAPO-34 as a function of polymer type, SAPO-34 dosage, pressure, and temperature. A hybrid optimization algorithm and sensitivity analysis determine this model’s adjustable coefficient and its hyperparameters, respectively. The proposed model in this study can help construct the MMM-based CO_2_ removal facility by providing accurate predictions for CO_2_ permeability as a function of MMM chemistry and operating conditions.

## 2. Literature Data

Different MMMs have already been fabricated by combining SAPO-34 zeolite and Matrimid-5218 [24], Pebax-1657 [25], polyethersulfone [26,27], polyurethane [28], polysulfone [29], and Pebax-1074 [30] polymers for the purpose of CO_2_ removal. The literature measured the CO_2_ permeability in these MMMs as a function of SAPO-34 dosage, temperature, and pressure. Table 1 presents numerical values of the statistical information (minimum, maximum, average, and standard deviation) of these experimental measurements, and Figure 1 illustrates their graphical version using the box plot. It can be seen that the collected experimental databank covers SAPO-34 dosage, temperature, and pressure ranges of 0–50 wt.%, 267–394 K, and 0.1–3 MPa. In this condition, the CO_2_ permeability varies from 0.2–337 barrer. 

In addition to these statistical and graphical presentations of the experimental measurements, the Appendix A reports all their information.

## 3. ANFIS Description

### 3.1. ANFIS Structure

The function approximation [31] and classification [32] are the problems that can be easily handled by the machine learning tools. Therefore, this study aims to efficiently tune the adaptive neuro-fuzzy inference system topology to enhance the prediction accuracy of CO_2_ permeability in different MMMs in a broad range of operating conditions. Consequently, it seems necessary to briefly describe the working procedure of this intelligent estimation machine.

The ANFIS is a powerful modeling tool that simultaneously utilizes the inference property of fuzzy logic and the learning capabilities of artificial neural networks. Consider a multiple-input and single-output (MISO) regression problem, such as *F* (*X*, *Y*). A Takagi–Sugeno system with five interconnected successive layers (i.e., fuzzification, interference, normalization, interpolation, and target computation) and the following fuzzy if–then rules, namely Equations (1) and (2) [33], can be constructed to model this problem.
Rule 1: If X is A_1_ and Y is B_1_, then F_1_ = C_1_ × X + D_1_ × Y + E_1_(1)
Rule 2: If X is A_2_ and Y is B_2_, then F_2_ = C_2_ × X + D_2_ × Y + E_2_(2)

Here, A_1_, A_2_, B_1_, and B_2_ are the premise parameters. On the other hand, the adjustable consequence parameters are shown by C_1_, C_2_, D_1_, D_2_, E_1_, and E_2_.

The first layer assigns a membership function (η) to each node j and calculates the output signals ($O_{j}^{1}$) based on Equations (3) and (4) [34].
(3)$$O_{j}^{1}\;=\;\eta_{A_{j}}\left(X\right)\;\;\;\;\;\;j=1,\;2$$
(4)$$O_{j}^{1}\;=\;\eta_{B_{j-2}}\left(Y\right)\;\;\;\;\;\;j=3,\;4$$

The output of the second layer ($O_{j}^{2}$) is the product of all the incoming signals [35].
(5)$$O_{j}^{2}\;=\;\eta_{A_{j}}\left(X\right)\;\eta_{B_{j}}\left(Y\right)\;=\;\omega_{j}\;\;\;\;\;\;j=1,\;2$$

The outputs of the third layer ($O_{j}^{3}$) compute by normalizing the entry signals, as in Equation (6) [36].
(6)$$O_{j}^{3}\;=\;\bar{\omega_{j}}\;=\;\omega_{j}/\left(\omega_{1}\;+\;\omega_{2}\right)\;\;\;\;j=1,\;2$$

Then, Equation (7) calculates the outputs of the fourth layer ($O_{j}^{4}$) [37].
(7)$$O_{j}^{4}\;=\;\bar{\omega_{j}}\;F_{j}=\;\bar{\omega_{j}}\;\left(C_{j}\;X\;+\;D_{j}\;Y\;+\;E_{j}\right)\;\;\;\;j=1,\;2$$

Finally, the ANFIS prediction for the target ($O^{ANFIS}$) can be achieved by applying Equation (8) [38].
(8)$$O^{ANFIS}\;=\;\sum_{j=1}^{2}\bar{\omega_{j}}\;F_{j}$$

### 3.2. Input Space Partitioning Strategies

The literature declares that the fuzzification strategy (the first layer of ANFIS) significantly impacts the accuracy of simulating a given problem [39]. Generally, there are three well-known methods to divide the input space, i.e., grid partitioning (GP), fuzzy C-means (FCM), and subtractive clustering (SC). Although Yeom and Kwak comprehensively described GP, FCM, and SC and highlighted their weaknesses and strengths [33], the following subsections briefly explain these partitioning strategies.

#### 3.2.1. Grid Partitioning

Grid partitioning is a method that breaks down input space into a non-overlapping grid-like structure. This method is suitable for a problem with a small number of independent variables in low-dimensional input space. For instance, 1024 unique areas and a rule for each area (i.e., 1024 rules) have been provided by the GP method with two membership functions for a problem with 10 independent variables [33]. Hence, the GP is a complicated strategy to divide the input space of a high-dimensional problem (high number of the independent variable of observations).

#### 3.2.2. Fuzzy c-Means Clustering

The FCM separates the input space and places them in different fuzzy clusters with a specific radius. An optimization method that minimizes the observed non-similarity determines these properties. The user specifies the number of fuzzy rules by setting the number of clusters. 

#### 3.2.3. Subtractive Clustering

The SC is a scenario to break down the input space into several specific clusters by performing a multi-dimensional analysis. This strategy automatically divides the input space into an appropriate number of clusters utilizing a user-entered value for the cluster radius. The SC generates a high number of small-sized clusters when a small cluster radius is entered, and vice versa.

### 3.3. Training Algorithm

Traditionally the backpropagation or hybrid training algorithm is applied to determine adjustable parameters of membership functions during the training step of the ANFIS. The backpropagation method continuously updates the adjustable parameters of the ANFIS to minimize the deviation between actual and predicted values of a dependent variable. Indeed, the observed deviation propagates backward through the ANFIS structure, and its parameters adjust by several iterations. On the other hand, the hybrid method combines the least-squares and backpropagation algorithms to tune ANFIS parameters efficiently. To enhance the performance of the training phase, as well as to avoid errors related to the different ranges of variables, the experimental data are needed to normalize between 0 and 1 [40,41]. 

## 4. Results and Discussion

The most appropriate structural features for three well-known ANFIS types have been determined in this section. Four statistical uncertainty criteria, i.e., absolute average relative deviation (*AARD*%), mean squared error (*MSE*), root mean squared error (*RMSE*), and correlation of determination (*R*), are often applied to perform such comparison [42,43]. Equations (9)–(12) express that these uncertainty criteria quantize the compatibility between laboratory-measured CO_2_ permeability in MMMs ($\lambda_{CO_{2}}^{lab}$) and its counterpart simulation by the ANFIS models ($\lambda_{CO_{2}}^{sim}$).
(9)$$R=\sqrt{1-\;\left\{\sum_{i=1}^{n}{\left(\lambda_{CO_{2}}^{lab}-\lambda_{CO_{2}}^{sim}\right)}_{i}^{2}/\sum_{i=1}^{n}{\left(\lambda_{CO_{2}}^{lab}-\bar{\lambda_{CO_{2}}^{lab}}\right)}_{i}^{2}\right\}}$$
(10)$$AARD\%=\;\left(100/n\right)\;\;\times \;{\sum_{i=1}^{n}\left(\left|\lambda_{CO_{2}}^{lab}-\lambda_{CO_{2}}^{sim}\right|/\lambda_{CO_{2}}^{lab}\right)}_{i}$$
(11)$$MSE=\;\left(1/n\right)\;\;\times \;{\sum_{i=1}^{n}\left(\lambda_{CO_{2}}^{lab}-\lambda_{CO_{2}}^{sim}\right)}_{i}^{2}$$
(12)$$RMSE=\;\sqrt{\left(1/n\right)\;\;\times \;{\sum_{i=1}^{n}\left(\lambda_{CO_{2}}^{lab}-\lambda_{CO_{2}}^{sim}\right)}_{i}^{2}}$$

Then, the most accurate neuro-fuzzy model is identified by the spider plot analysis. Different graphical analyses then deeply assess the performance of the selected ANFIS model. Finally, the effect of filler dosage, pressure, and temperature on the CO_2_ permeability in the MMM is investigated.

### 4.1. Developing ANFIS Models

As noted, GP, FCM, and SC strategies have been used to divide the input space of ANFIS. Therefore, it is possible to develop and compare three ANFIS types, i.e., ANFIS-GP, ANFIS-FCM, and ANFIS-SC. Table 2 reports the investigated ranges of structural features for each ANFIS type and the selected one by statistical analyses. This table states that the ANFIS-GP and ANFIS-FCM should construct with two and eleven clusters, respectively, while the cluster radius of the ANFIS-SC model should fix at 0.55. The hybrid algorithm is better at performing the training step of all three ANFIS types than the backpropagation.

Table 3 utilizes AARD%, MSE, RMSE, and R-value to measure/compare the accuracy of the selected ANFIS models. The accuracy has been separately monitored for the training, testing, and overall databank. Although all these accuracies are valuable from the modeling point of view, the performance of the ANFIS-FCM and ANFIS-SC is far better than the ANFIS-GP. The next sections apply the spider plot to select the most suitable ANFIS type to estimate CO_2_ permeability in the mixed matrix membranes containing SAPO-34 zeolite.

### 4.2. Choosing the Best ANFIS Type

Before performing the model selection analysis, it is better to highlight that a model with a low AARD%, MSE, RMSE, and close R-value to one is preferable from a modeling point of view. The spider plot is a graphical method to compare the statistical indexes provided by ANFIS-GP, ANFIS-FCM, and ANFIS-SC at a glance. In other words, the spider analysis presents the reported statistical values in Table 3 in a multi-dimensional graph.

The results of comparing the performance of ANFIS-GP, ANFIS-FCM, and ANFIS-SC in the training and testing groups and the whole database have been shown in Figure 2a–c, respectively. These spider graphs can be easily analyzed to compare the provided AARD%, MSE, RMSE, and R indexes by different ANFIS types.

The spider graph analyses (Figure 2a,b) indicated that the ANFIS-SC performs better when predicting the training and testing datasets of CO_2_ permeability in the MMMs. Since the overall databank combines the training and testing groups, it is obvious that the ANFIS-SC accuracy is also better than the two other models (Figure 2c). 

### 4.3. Evaluating the Performance of the Selected ANFIS Model

This section comprehensively evaluates the reliability of ANFIS-SC performance in predicting CO_2_ permeability in MMMs by combining statistical and visual inspections. The performance assessment has been separately performed for the training and testing groups and their combination (i.e., overall databank).

#### 4.3.1. Training Step

The correlation between experimental CO_2_ permeabilities in MMMs and their corresponding predictions by ANFIS-SC in the training step has been plotted in Figure 3a. Since all square symbols are gathered around the diagonal line, it can be claimed that the ANFIS-SC model accurately predicts the training datasets.

Visually comparing the actual and predicted values of a dependent variable is a well-known method to check the accuracy of a model [44]. Thus, numerical values of the CO_2_ permeability in MMMs in the training step, as well as their associated ANFIS-SC predictions, have been illustrated in Figure 3b. This figure approves excellent compatibility between laboratory-measured and ANFIS-SC predictions in the training phase.

Indeed, the ANFIS-SC predicts 88 actual CO_2_ permeabilities in MMMs with the AARD = 1.85%, MSE = 5.59, RMSE = 2.37, and R = 0.9996.

#### 4.3.2. Testing Step

This analysis correlates the experimental CO_2_ permeabilities in MMMs and their related ANFIS-SC predictions in the testing step (Figure 4a). It can be seen that the model predictions have been precisely mapped on the experimentally measured data. Indeed, the ANFIS-SC predicts 15 actual CO_2_ permeabilities in MMMs with the AARD = 4.38%, MSE = 70.70, RMSE = 8.41, and R = 0.9952.

Figure 4b also presents the numerical values of actual and estimated CO_2_ permeability in MMMs in the testing step of the ANFIS-SC. Since the model has seen none of these samples before, its excellent performance for simulating CO_2_ separation by the MMM can be observed. Indeed, the ANFIS-SC accurately predicts all unknown CO_2_ permeability in MMMs at the testing stage except for one sample.

#### 4.3.3. All Experimental Data

This section analyzes the ANFIS-SC performance in predicting the overall databank of CO_2_ permeability in mixed matrix membranes. Since the overall databank is a combination of the training and testing groups, it is expected that the ANFIS-SC model will also have an excellent performance in this analysis.

The experimental and simulated CO_2_ permeabilities in MMMs for training and testing groups (overall databank) have been presented in Figure 5a. This figure confirms a remarkable agreement between the laboratory-measured and simulated values. Indeed, the ANFIS-SC predicts 103 actual CO_2_ permeabilities in MMMs with the AARD = 2.22%, MSE = 15.08, RMSE = 3.88, and R = 0.9989.

Figure 5b also presents the ANFIS-SC prediction for each experimental measurement separately. The outstanding performance of this model in simulating CO_2_ separation by the mixed matrix membranes can be justified by this figure.

### 4.4. Investigating the Effect of SAPO-34 Dosage, Pressure, and Temperature 

The effect of filler dosage on the isothermal CO_2_ permeability (298 K) in Pebax-1074/SAPO-34 MMMs at six pressure levels has been illustrated in Figure 6. This figure displays experimental CO_2_ permeability profiles and their corresponding simulations by the ANFIS-SC model. The designed model not only correctly anticipates the trend of experimental profiles but also accurately estimates all individual data points. Actually, the proposed neuro-fuzzy model easily identifies the effect of pressure and filler dosage on the CO_2_ permeability in Pebax-1074/SAPO-34 mixed matrix membranes.

This figure also shows that the CO_2_ permeability in the Pebax-1074/SAPO-34 is enhanced by increasing either pressure or filler dosage. The maximum CO_2_ permeability of 250 barrer is achieved at the maximum pressure and SAPO-34 dosage of 2.4 MPa and 30 wt.%, respectively. 

The simulation and experimental profiles for the isobaric CO_2_ permeability (0.7 MPa) in Pebax-1657/SAPO-34 MMMs as a function of temperature have been shown in Figure 7. This figure can readily demonstrate the outstanding compatibility between the actual and simulation profiles. This analysis also indicates that the CO_2_ permeability in the pure Pebax-1657 may be increased by more than 40 barrers by adding 30 wt.% of SAPO-34 zeolite. In addition, the increasing effect of temperature on the CO_2_ permeability in both pure Pebax-1657 and Pebax-1657/SAPO-34 MMM is observable in Figure 7. 

### 4.5. Identifying Undesirable Outliers

The leverage method provides a practical ground to identify undesirable outliers in an experimental database [45]. Figure 8 shows that a small number of experimental data (i.e., 4 samples) are undesirable outliers. Since almost all the samples are valid, the undesirable effect of outliers on the model prediction can be ignored.

## 5. Conclusions

The current research combined the statistical and graphical analyses to compare the performance of three neuro-fuzzy types (i.e., ANFIS-GP, ANFIS-FCM, and ANFIS-SC) in predicting the CO_2_ separation ability of six different mixed matrix membranes containing SAPO-34 zeolite as a filler. The ANFIS-SC trained by the hybrid algorithm has been identified as the most reliable model for estimating the considered matter. This model predicted 103 experimentally measured CO_2_ permeability values in a wide range of membrane compositions and operating conditions with the AARD = 2.22%, MSE = 15.08, RMSE = 3.88, and R = 0.9989. In addition, the visual inspections have also justified the outstanding ability of the proposed neuro-fuzzy model in simulating CO_2_ separation by mixed matrix membranes. The simulation and experimental results showed that the CO_2_ permeability in all investigated MMMs can be increased by increasing the SAPO-34 dosage, pressure, and temperature. The proposed intelligent model in this study can quickly determine the membrane composition and operating conditions so that the CO_2_ permeability is maximized.

## Figures and Tables

**Figure 1 membranes-12-01147-f001:**
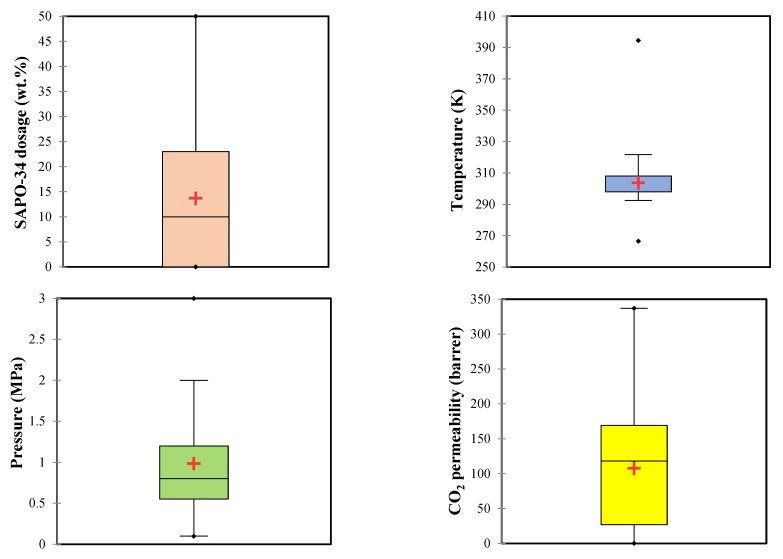
Box plot presentation of statistical information of the collected experimental databank.

**Figure 2 membranes-12-01147-f002:**
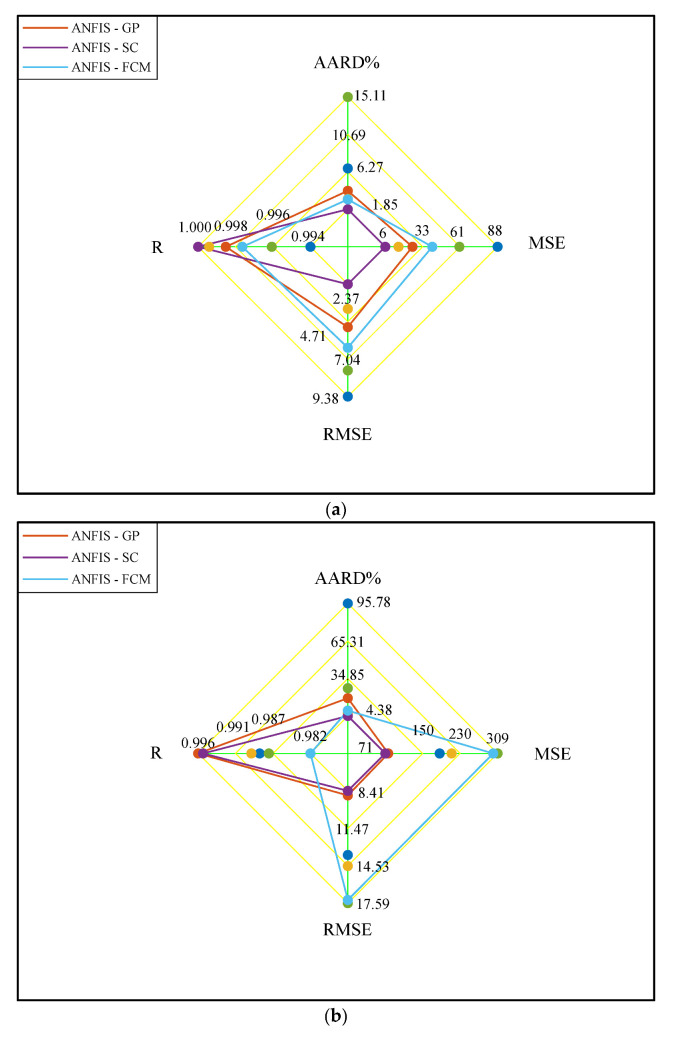
Spider plot for comparing the ANFIS models’ performance in the (**a**) training, (**b**) testing, and (**c**) all of the database.

**Figure 3 membranes-12-01147-f003:**
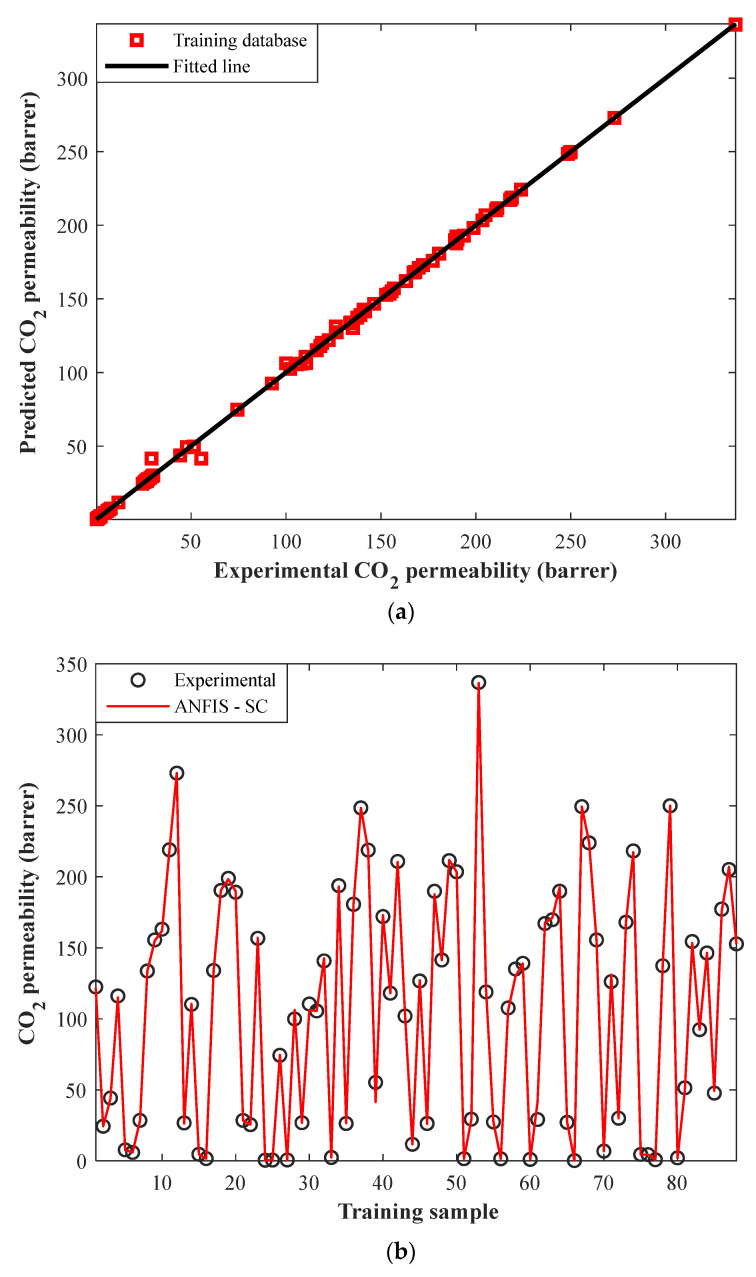
(**a**) Cross-plot of ANFIS-SC predictions versus experimental CO_2_ permeabilities in the training step. (**b**) Experimental CO_2_ permeabilities and their related ANFIS–SC predictions in the training step.

**Figure 4 membranes-12-01147-f004:**
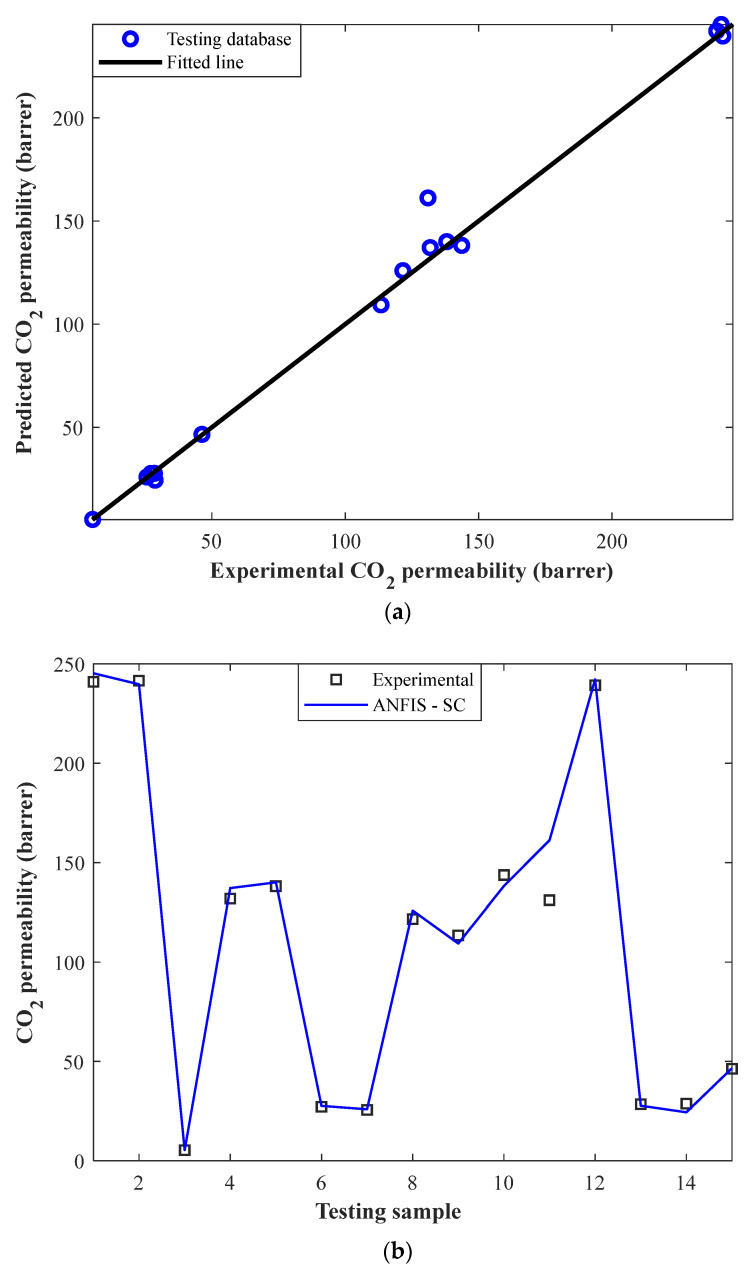
(**a**) Cross-plot of ANFIS-SC predictions versus experimental CO_2_ permeabilities in the testing step. (**b**) Experimental CO_2_ permeabilities and their related ANFIS–SC predictions in the testing step.

**Figure 5 membranes-12-01147-f005:**
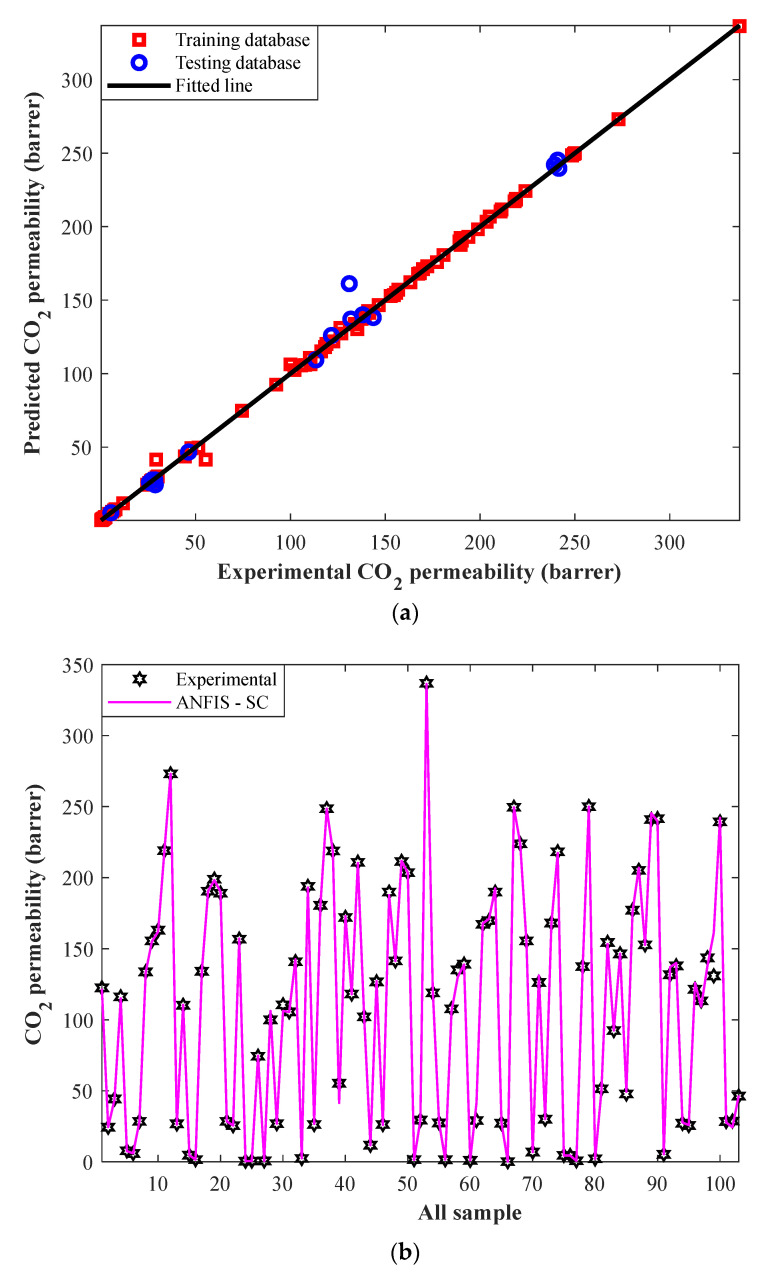
(**a**) Correlation between experimental and predicted CO_2_ permeabilities for the whole of experimental databank. (**b**) Experimental CO_2_ permeabilities and their associated ANFIS-SC predictions for the combination of training and testing datasets.

**Figure 6 membranes-12-01147-f006:**
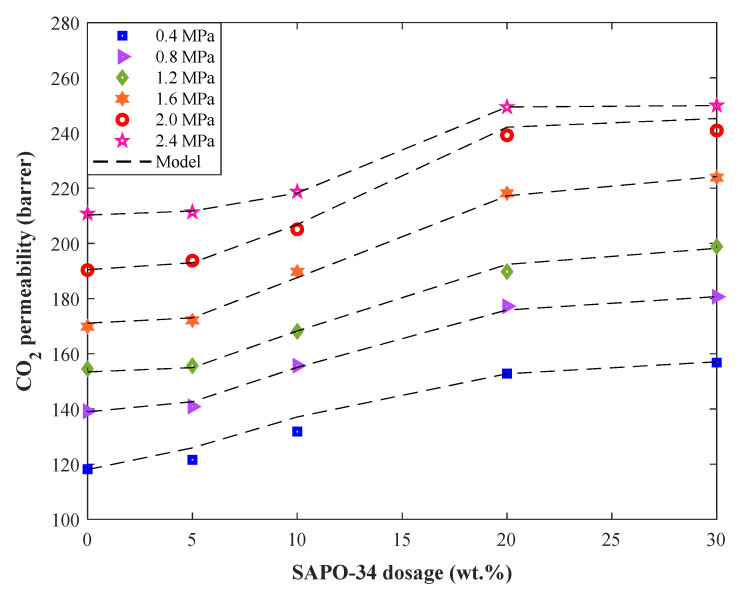
Monitoring the effect of SAPO-34 dosage and pressure on the CO_2_ permeability in Pebax-1074-based MMM (298 K).

**Figure 7 membranes-12-01147-f007:**
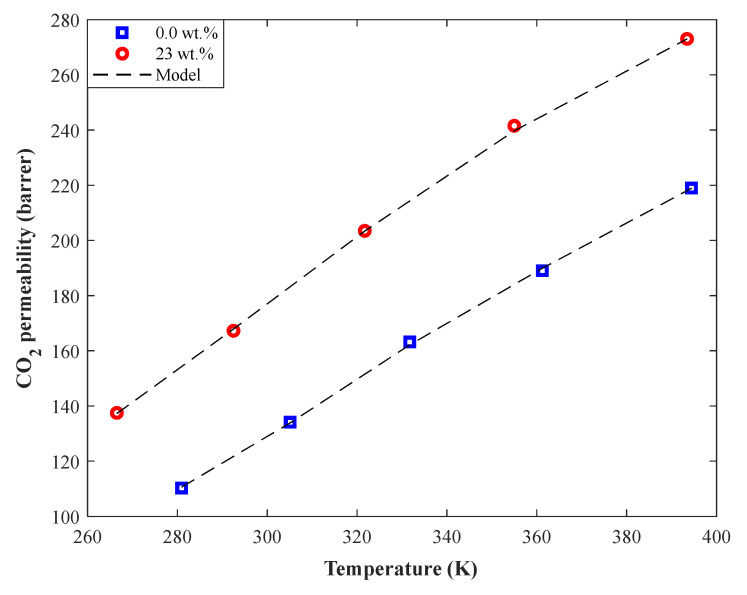
The effect of temperature on the CO_2_ permeability in pure Pebax-1657 and Pebax-1657/SAPO-34 MMM (0.7 MPa).

**Figure 8 membranes-12-01147-f008:**
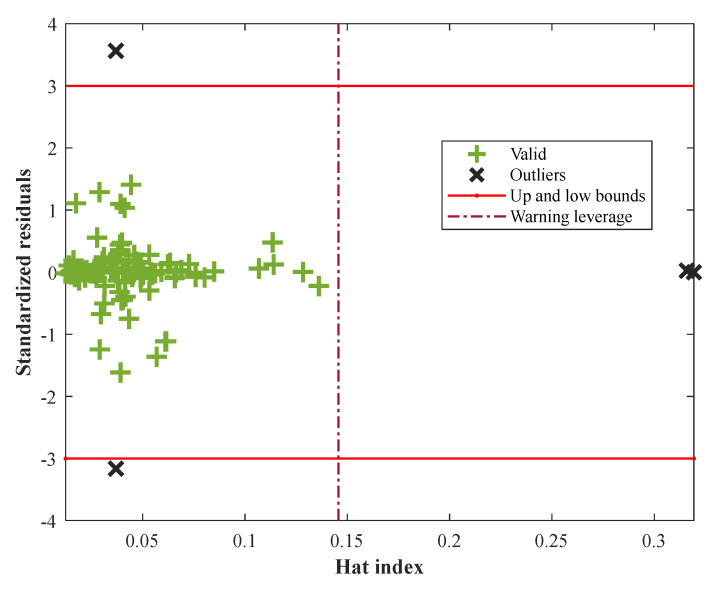
Employing the leverage method to identify undesirable outliers.

**Table 1 membranes-12-01147-t001:** Statistical information of the collected experimental databank [24,25,26,27,28,29,30].

Variable	Min	Max	Average	Standard Deviation
SAPO-34 dosage (wt%)	0	50	13.70	11.75
Temperature (K)	267	394	303.75	16.57
Pressure (MPa)	0.1	3	0.985	0.614
CO_2_ perm (barrer)	0.20	337	107.88	83.64

**Table 2 membranes-12-01147-t002:** The checked and selected structural features of different ANFIS types.

Model	Checked Features	The Best Features
ANFIS-GP	Cluster number: 2:1:5 Optimization algorithm: backpropagation and hybrid	Two clusters Hybrid
ANFIS-FCM	Cluster number: 2:1:12 Optimization algorithm: backpropagation and hybrid	Eleven clusters Hybrid
ANFIS-SC	Cluster radius: 0.1:0.05:1 Optimization algorithm: backpropagation and hybrid	Cluster radius = 0.55 Hybrid

**Table 3 membranes-12-01147-t003:** Performance analysis of the selected ANFIS models by four statistical indexes.

Model	Database	AARD%	MSE	RMSE	R
ANFIS-GP	Training	4.03	25.54	5.05	0.9981
	Testing	18.91	77.05	8.78	0.9958
	Overall	6.19	33.05	5.75	0.9976
ANFIS-FCM	Training	3.02	40.09	6.33	0.9972
	Testing	8.78	299.98	17.32	0.9823
	Overall	3.86	77.94	8.83	0.9946
ANFIS-SC	Training	1.85	5.59	2.37	0.9996
	Testing	4.38	70.70	8.41	0.9952
	Overall	2.22	15.08	3.88	0.9989

## Data Availability

The analyzed experimental data in this research is available in the Appendix A.

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
