# Peer review of "Developing a Hybrid Neuro-Fuzzy Method to Predict Carbon Dioxide (CO2) Permeability in Mixed Matrix Membranes Containing SAPO-34 Zeolite"

_membranes, 2022, doi:10.3390/membranes12111147_

Round 1
Reviewer 1 Report
1. In Table 1, the author did not indicate the source where the data is obtained. Furthermore, with the range of 0.20 to 337 barrer (The unit "barrer" should be spelt in this form and not "Barrer") is too wide, and it will affects the accuracy of the result. The author may need to read the literature with care and remove the undesirable outliers.
2. What polymer matrices are used in this study? There is no such information in Table 1.
3. The equation formatting is very untidy, and it is difficult for the readers to understand properly.
4. Why for Figure 7, there is no data at intermediate loadings, as compared to Figure 6?
Author Response
Comments and Suggestions for Authors
Dear Reviewer 1
We would like to thank you for your constructive comments and the time put into reviewing the manuscript. We believe the manuscript has substantially improved after carefully addressing your valuable comments and making the suggested modifications. The corresponding changes are summarized in our response below and highlighted in yellow in the revised manuscript.
We hope our revisions have improved the paper quality to your satisfaction level.
- In Table 1, the author did not indicate the source where the data is obtained. Furthermore, with the range of 0.20 to 337 barrer (The unit "barrer" should be spelt in this form and not "Barrer") is too wide, and it will affects the accuracy of the result. The author may need to read the literature with care and remove the undesirable outliers.
Response: The following actions are made to address these valuable comments:
A1 – The references of the experimental data have been added to the revised manuscript (Line 87)
A2 – The spell of all “Barrer” terms in text, tables, and figures have been changed to “barrer” (highlighted by green)
A3 –The numerical value of accuracy indexes (Table 3) and visual inspection of figures 2-7 approve that the developed model precisely predicts the actual CO2 permeability data in this wide range. Indeed, this observation is an indication of excellent generalization ability of the hybrid neuro-fuzzy model.
A4 – The revised manuscript utilizes the leverage method to identify the undesirable outliers. A little number of the experimental data has been identified as an outlier (Lines 287-293).
- What polymer matrices are used in this study? There is no such information in Table 1.
Response: The information of polymer matrices has been presented in the revised manuscript (Lines 77-79). In addition, all experimental measurements have been added to the revised manuscript as a supplementary file.
- The equation formatting is very untidy, and it is difficult for the readers to understand properly.
Response: We did our best to modify the equation formatting in the revision stage.
- Why for Figure 7, there is no data at intermediate loadings, as compared to Figure 6?
Response: This figure aims to analyze the effect of temperature on the CO2 permeability. In addition, upon checking the experimental data in the supplementary file, you can see that:
B1 – The CO2 permeability as a function of the temperature is only available for the Pebax-1657 and Pebax-1657/SAPO-34 MMM
B2 – There is only two zeolite loading levels for this polymer matrix.
Reviewer 2 Report
The manuscript compares the predictive performance of three neuro-fuzzy types (ANFIS) in predicting the permeability of carbon dioxide (CO2) in mixed matrix membrane (MMM) containing the SAPO-34 zeolite. This model predicted 103 experimentally measured CO2 permeability in a wide range of membrane compositions and operating conditions with the AARD=2.22%, MSE=15.08, RMSE=3.88, and 287 R=0.9989. Based on these results, the model proposed in this manuscript is very useful to provide reference for selection of membrane materials in gas separation fields. Overall, the research subject is important. The idea is great and experiments carried out are comprehensive. The article is well written and could be publishable in Membranes
Author Response
The manuscript compares the predictive performance of three neuro-fuzzy types (ANFIS) in predicting the permeability of carbon dioxide (CO2) in mixed matrix membrane (MMM) containing the SAPO-34 zeolite. This model predicted 103 experimentally measured CO2 permeability in a wide range of membrane compositions and operating conditions with the AARD=2.22%, MSE=15.08, RMSE=3.88, and 287 R=0.9989. Based on these results, the model proposed in this manuscript is very useful to provide reference for selection of membrane materials in gas separation fields. Overall, the research subject is important. The idea is great and experiments carried out are comprehensive. The article is well written and could be publishable in Membranes.
Dear Reviewer 2
Thank you so much for suggesting the acceptance of this manuscript in the Membranes journal.
Round 2
Reviewer 1 Report
It can be published as it is.